

# Criteria-based visualization design for hazard maps

Max Schneider[1,2], Fabrice Cotton[1,3], and Pia-Johanna Schweizer[2]

[1]German Research Centre for Geosciences, Potsdam, Germany
[2]Institute for Advanced Sustainability Studies, Potsdam, Germany
[3]University of Potsdam, Germany

**Correspondence:** Max Schneider (maxs@gfz-potsdam.de)

**Abstract.** Probabilistic seismic hazard estimates are a key ingredient of earthquake risk mitigation strategies and are often communicated through seismic hazard maps. Though the literature suggests that visual design properties are key for effective communication using such maps, guidelines on how to optimally design hazard map are missing from the literature. Current maps use color palettes and data classification schemes which have well-reported limitations that may inadvertently miscom-
municate seismic hazard. We surveyed the literature on color and classification schemes to identify design criteria that have empirical support for communicating hazard information. These criteria were then applied to redesign the seismic hazard map for Germany. We isolated several communication goals for this map, including essential properties about moderate-hazard seismic regions and a critical hazard threshold related to the German seismic building codes. We elucidate our redesign process and the selection of new colors and classification schemes that satisfy the evidence-based criteria. In a mixed-methods
survey, we evaluate the original and redesigned seismic hazard maps, finding that the redesign satisfies all the communication goals, and improves users' awareness about the spatial spread of seismic hazard, relative to the original. We consider practical implications for the design of hazard maps across the natural hazards.

## 1 Introduction

### 1.1 Probabilistic seismic hazard analysis and maps

Earthquakes are social problems because they cause ground shaking that can destroy buildings and injure or kill people. Ground shaking due to earthquakes is referred to as seismic hazard and physical models can assess the amount of ground shaking that can be expected at any given location in a seismic region. These models are functions of parameters that describe both how frequently various earthquakes occur in the region, and the full range of ground shaking that each earthquake may result in.

In a modern probabilistic seismic hazard assessment (PSHA) aiming to capture epistemic uncertainties, parameters for these
models are varied systematically to create a hazard ensemble over many model runs. The probabilistic distribution of seismic hazard values across a region is highly skewed right, owing to the low probability of high shaking; this is because larger-magnitude earthquakes (which generate higher levels of shaking) have a correspondingly lower probability of occurring. In moderate-seismicity regions, the majority of the region may have hazard areas mapped as low, with few zones of higher hazard and fewer zones still of extreme hazard. These unlikely but extreme hazard levels have a strong impact on the expected losses,





in particular in urban areas. Still, damaging shaking from earthquakes may also occur (of course, with a low probability) where
hazard is mapped as low.

Probabilistic hazard assessments commonly show the horizontal ground acceleration (peak ground acceleration, PGA) which
is exceeded, "on average", every 475 years (e.g., Baker et al. (2021)). This so-called "475-year return period" means that, due
to the assumption that earthquakes follow a Poisson distribution, this acceleration has a 10% probability of being exceeded in

50 years. Because earthquakes are generally clustered into certain zones in a region (e.g., around fault lines), hazard varies over
space and PSHA ensembles are produced over fine-scale grids across a region. The resulting seismic hazard maps depict the
spatial variation of a summary, e.g., the mean, of the ensemble of hazard outcomes. These hazard maps are used by a variety of
user groups, from political or economic decision-makers assessing a seismic region's land use to civil engineers and officials
responsible for creating seismic-resistant buildings. Urban seismic risk is of particular importance, as highly developed urban

areas have greater exposure to earthquake damages.

### 1.1.1   German seismic hazard map

In our study, we consider seismic hazard mapping for the country of Germany. The seismicity of Germany is elevated in certain
regions of the country, when compared to other parts of central Europe, particularly in the southwest and along the Rhine River.
In general, the seismicity is indeed low in relation to the plate-boundary regions of the Mediterranean (Grünthal et al., 2018).

Still, "no part can be regarded as aseismic" (Tyagunov et al., 2006), i.e. damaging seismic events can be expected, in principle,
everywhere. The national German seismic hazard map is shown in Fig. 1 (reproduced from Fig. 28 (middle) in Grünthal et al.
(2018)). This shows the mean PGA value that has a 10% probability exceedance in 50 years, with the mean taken over 4040
model runs in the probabilistic ensemble. The PSHA described herein was accomplished on behalf of the Deutsches Institut für
Bautechnik (German Institute for Civil Engineering) and was launched by the respective national committee on standardization

of the Deutsches Institut für Normung (German Institute for Standardization).

The map in Fig. 1 was produced following consultations with users of PSHA products (Grünthal, 2021) but, as discussed in
the subsequent sections, this map has some disadvantages. It uses a rainbow color palette, which is known to be suboptimal for
visual perception, with reported psychological associations that are not appropriate for seismic hazard, and less accessible to
users with color vision deficiency. The legend associates these colors with hazard levels in a way that neither groups together

hazard levels that are closer together, nor spotlights key hazard thresholds. The legend further extends to a much higher value
than what is shown in this map, likely because the same legend is used to map other (higher) parts of the PSHA ensemble (see
Sect. 2.2). In this paper, we solely consider single-map representations of hazard and what the research evidence supports to
improve such maps.

Visualization of seismic hazard can be challenging due to its spatial variation and highly skewed distributions, but these

issues are ubiquitous across earthquake data (Schneider et al., 2022; Bostrom et al., 2008). As a starting point for our framework
for hazard map design, we specified what the viewer should ideally understand after reading the German seismic hazard map.
We argue that these properties are fundamental to the understanding of how seismic hazard changes across a region, for the



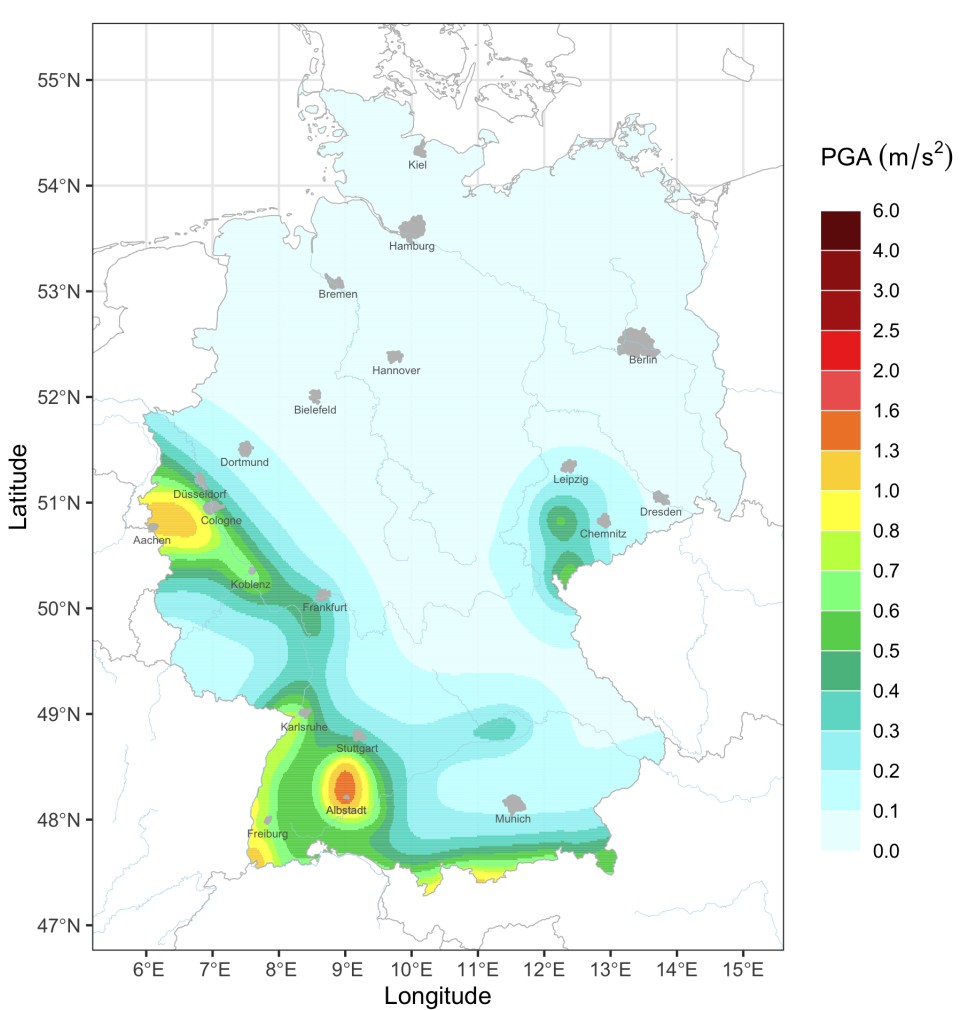

**Figure 1.** National seismic hazard map for Germany, reproduced from Fig. 28 (middle) in Grünthal et al. (2018). It shows the peak ground acceleration (PGA, in $\frac{m}{s^2}$) expected at each location for a return period of 475 years, taking the mean over the probabilistic ensemble.



largest variety of users. The communication goals for the map include the following four properties about the seismic hazard phenomenon:

1. Probabilistic seismic hazard (shaking) is not spotty but rather, changes continuously in space because of the progressive decay of seismic waves generated by earthquakes and the fact that earthquakes may occur over large areas outside well-known faults.

   2. Earthquakes and associated damaging shaking may occur even in areas where no earthquakes have historically been observed and therefore where seismic shaking is rare and (probabilistic) seismic hazard is mapped as low. As such, there
is no zero-hazard area on the map.

   3. The highest end of the hazard distribution has a large impact on expected losses and extreme hazard thus differs from the preceding levels of high hazard.

   4. Spatial patterns of hazard may be more important (e.g., for situational awareness) than the specific magnitude of the hazard.

**1.2   Visual communication for hazard maps**

Beyond identifying communication goals, mapping seismic hazard requires many practical choices on how to best represent the hazard values on a map. Though the amount of literature on the visual communication of seismic hazard is far smaller than on the models thereof, previous scholars have investigated how to best map seismic hazard.

Gaspar-Escribano and Iturrioz (2011) provide a series of general guidelines they call "experience-based" regarding which
hazard parameters to map and how to select colors and graphical elements for seismic hazard maps. These issues are also highlighted in Fyfe and Molnar (2020), who investigate candidate seismic hazard maps for the Vancouver, Canada metro area with a stakeholder workshop and survey. The authors identify three key factors for effective visual communication: (1) choosing the right colors and graphical properties; (2) choosing appropriate hazard metrics and classifying them into discrete intervals for the map; and (3) using accessible data formats. Similar work on flood and volcano hazard communication has designed
hazard maps and evaluated them with focus groups and interviews (Hagemeier-Klose and Wagner, 2009; Meyer et al., 2012; Haynes et al., 2007).

Complementary to these studies, Marti et al. (2019) use a large-sample online survey to study users' preferences and ability to read information off maps presenting different seismic hazard information for Switzerland. They find that maps showing hazard via maximum expected earthquake magnitudes or intensities are less successful than traditional hazard maps showing, e.g.,
PGA. They conclude that this is because those alternate maps fail to follow best practices related to color and data classification, but do not explain precisely what would improve them. Thompson et al. (2015) use both semi-structured interviews and a survey with stakeholders to test different designs for volcanic hazard maps in New Zealand. They find that participants both prefer and have higher map-reading accuracy with maps where data is classified into discrete intervals, and that changing the color palette can change the hazard level perceived for a given location, as also found for tornado hazard maps by Miran et al. (2017).



A common finding across previous research is the importance of colors and data classification in how hazard maps are used and interpreted. But authors omit specifics of how and why to make particular design choices, typically citing so-called "best practices" used in their hazard maps without evidence (Hagemeier-Klose and Wagner, 2009; Meyer et al., 2012; Fyfe and Molnar, 2020; Marti et al., 2019). Indeed, Thompson et al. (2015) summarize the state-of-the-art for hazard map design: "fundamental choices for color scheme ... and data classification are largely driven by subjective preference." The problem is

that suboptimal choices for either aspect may lead to distortions in how hazard is understood, especially by non-scientific users (Evans, 1977; Ware et al., 2018; Dasgupta et al., 2018). There is a research gap in identifying the key criteria that should guide these essential decisions in designing a hazard map.

### 1.3  Criteria-based hazard mapping

A vast body of literature exists on color palettes and data classification schemes, both generally and specifically for natu-

ral hazards. Theoretical work has provided frameworks for considering both map elements. For example, color models can parametrize all colors into district dimensions, which can help isolate which aspects of color may be most critical for a hazard map. Similarly, scholars in cartography have proposed various methods of classifying data to produce discrete maps, to theoretically optimize how data is grouped for maps.

Research in these fields have further studied what empirical effects different design decisions have on how well users can

read and use the corresponding visualizations (Bostrom et al., 2008; Kinkeldey et al., 2017). In controlled experiments, users perform tasks that elucidate whether or not a design choice can produce maps that are not merely properly read, but also properly understood. Such empirical evaluations can estimate the effects that a given design choice has on map-reading and perception. We use this literature to isolate which criteria can facilitate effective colors and data classification for seismic hazard maps.

We apply these criteria to redesign the German seismic hazard map in Fig. 1. Since several candidate maps are consistent with the color and classification criteria, we decide between them using the four target properties of hazard we seek to communicate (see Sect. 1.1.1). We thus heed the call of previous authors to design maps that satisfy clearly specified communication goals (Dransch et al., 2010; Thompson et al., 2017). We then rigorously evaluate our redesigned map against the baseline map to investigate how well users can read and understand seismic hazard from the two maps. In detailing our design and evaluation

process, we attempt to provide a blueprint for future hazard map designers.

### 1.4  Research aims and contributions

This work seeks to fill the research gap in systematizing how to select colors, classification schemes, and other key features for hazard maps. We collect evidence-based practices from the color, visualization and cartographical literatures, identifying criteria that are supported by empirical experiments. Our proposed method for hazard mapping is centered around not only

following research-backed design criteria, but also in meeting communication targets about the hazard phenomenon.

We showcase this method with a case study for the German seismic hazard map. We redesign the original hazard map, following the specified criteria and making design choices to highlight the key properties of seismic shaking we wish to





communicate. We then evaluate the redesigned map with a controlled mixed-methods survey, using questions for not only map-reading but also hazard perception, which should be a primary objective of visual communication Dransch et al. (2010); MacPherson-Krutsky et al. (2020). The perception questions in our evaluation survey are directly linked to the key properties we wish to communicate.

The research questions we seek to address are as follows:

1. How can the design of seismic hazard maps be improved to both follow evidence-based practices and to aid the visual communication of key points about seismic hazard?

2. How well can a hazard map, redesigned for better visual communication, be read and interpreted by a technical audience? How does this compare to the previous (baseline) map?

## 2 Methods for selecting colors and classification schemes

### 2.1 Color: Literature review and best practices

Color is a critical visualization variable because "it is considered pre-attentive, meaning that information is extracted by the eye intuitively and rapidly" (Sherman-Morris et al., 2015). It is also an expressive and associative variable, with different colors producing different connotations (Itten, 1961). As such, care must be taken to select color palettes in hazard maps. There are well-evidenced criteria for selecting optimal colors for a palette and avoiding those colors that can lead to misinterpretations or unintended perceptions. For each criterion, we review the empirical literature that shows how ignoring it may endanger user understanding of data visualizations, including hazard maps.

To introduce key color issues, we use a color model, which expresses every possible color with a set of distinct variables (usually three). In the Hue-Saturation-Lightness (HSL) model, any color can be specified using values for Hue (0-360, representing the color's position on the color wheel), Saturation (0-100, where increasing values correspond to more intense and deep versions of the color) and Lightness (0-100, where increasing values correspond to lighter versions of the color, with 0 yielding pure black and 100 pure white). Many other color models exist that separate the color's luminance from the color's hue and its intensity, e.g., the Hue-Saturation-Brightness model and the CIE-LUV model (Zhou and Hansen, 2015). Although more sophisticated color models have certain benefits (e.g., the CIE-LUV model transforms the color space such that lightness increases in non-linear ways, which is closer to how the human eye perceives it), we use the simpler HSL model throughout this paper, as it is sufficient to describe the key color criteria we propose for hazard maps.

In an important study proposing a typology for color palettes, Bujack et al. (2017) posit that the perceptual order of the colors is a fundamental requirement when colors are to be compared, echoing many earlier authors (e.g., Tajima (1983); Trumbo (1981)). Colors may be ordered by their value on variables within the color model (for example, hue, lightness and saturation); it is desired for a palette's perceptual order based on any color variable to be the same as its order based on data value. It has been shown that changes in color lightness are the primary driver of human perception of color differences, with hue being a secondary driver (Spence et al., 1999; Kindlmann et al., 2002). Many authors (e.g., Thyng et al. (2016); Light



and Bartlein (2004)) have argued that a monotonic increase in the lightness of the colors is required for a perceptually-ordered color palette. Empirical experiments have found that unordered palettes in maps can lead to inaccurate map-reading (Dasgupta et al., 2018) and more difficulty in selecting between real map features (Rogowitz et al., 1999).

Another key aspect of a successful color palette is perceptual uniformity, in which unit increases in data value correspond to unit increases in the perception of change between colors, consistently across the entire color palette. To achieve such a uniform
perception of color differences throughout a map, authors have argued for not simply monotonic but linear progressions in color lightness across the entire palette (Robertson and O'Callaghan, 1986; Tajima, 1983). A palette that is not perceptually uniform can lead to perceptual distortions, with small data differences in one end of the palette being perceived differently than corresponding data differences at the other end (see Fig. 4 of Crameri et al. (2020)). Such a non-uniform reading of the colors was found in Ware et al. (2018), where participants had to detect background patterns in maps with color palettes that
either had linear progressions in lightness (perceptually uniform) or not. The authors found that in palettes where lightness changed non-linearly, participants detected the patterns worse for colors in the center of the palette, and especially when using the rainbow color palette.

A third critical dimension spotlighted by numerous authors is color discriminability (Bujack et al., 2017; Rheingans, 2000). For users to correctly interpret a color map, colors indicating important differences in data value should be separable from one
another, both in the map and its legend (Trumbo, 1981). Definitions for color discriminability have been proposed based on perceptual difference (i.e., difference in lightness or other color variables) (Maxwell, 2000; MacAdam, 1942) or how colors are identified by color names (Gramazio et al., 2016). Discriminable palettes have been associated with unit increases in both the hue and lightness profiles of a color palette (Trumbo, 1981). In a user experiment, Gramazio et al. (2016) found that palettes with optimized discriminability were superior to default palettes from oft-used software, as they often led participants
to more accurately identify which half of a map contained a reference color; participants also consistently preferred the palettes optimized for discriminability.

Colors should also be chosen that have psychological associations that befit the phenomenon being mapped. A body of empirical literature suggests that colors have consistent associations across individuals, which may be culture-specific (Wang et al., 2014). Typically, these studies ask participants to link a list of emotions or experiences with a set of colors. In many
Western cultures, blue is "often associated with openness, peace, and tranquility" (Mehta and Zhu, 2009) and blue and green have been found to have "the qualities of being comfortable and soothing" (Clarke and Costall, 2008). On the other hand, red hues have been found to be associated with dangers and risks within Western cultures, while yellows are associated with caution and warning (Griffith and Leonard, 1997), which has been found to persist across Western and Asian cultures (Or and Wang, 2014; Chan et al., 2003). In a user experiment, Lin et al. (2013) found quicker responses on a chart-reading task when
charts were colored using "semantically-resonant" colors that respected common associations rather than colors that did not, especially when the data being presented was more "colorable", i.e., had a larger degree of color associations.

Finally, colors should be chosen based on principles of accessibility and inclusion, including for those audience members with color vision deficiency (CVD), or the inability to differentiate between certain colors. About 8% of males of European Caucasian ancestry suffer from some CVD; in other populations, the prevalence is lower but still non-trivial (Birch, 2012).



The most common forms of CVD are deuteranomaly and protanomaly (red-green colorblindness) in which reds and greens of equal lightness are both seen as an identical shade of dark yellow (see Fig. S1 in the Supplement). People with this CVD can misread maps using the wrong color palettes. In Olson and Brewer (1997), participants with red-green blindness could read off colors and features in maps significantly better when using color palettes that were CVD-friendly (monotonically varying in lightness and avoiding red/green combinations) vs. -unfriendly (varying in hues and lightness with red and green together).

### 2.1.1   Criteria for selecting color palettes

The literature on which color aspects are critical for effective visual communication has not been lost on other geoscientific communities. Stauffer et al. (2015) discuss color concepts for heat maps common in meteorology, pinpointing the importance of differences in lightness across the palette and appropriate color conventions. Thyng et al. (2016) provides color guidelines and palettes for oceanography, spotlighting perceptual uniformity, cultural implications, and colorblind-friendliness as key criteria. Horton et al. (2020) proposes a "perception-informed color palette" for heat maps for avalanche science, optimizing it to be accessible for various CVDs. Expanding on this literature, we propose the following five critical criteria for consideration when choosing colors for hazard maps. While we do not claim these criteria are either sufficient or minimal for hazard map design, they align with lists and typologies published elsewhere; see e.g., Thyng et al. (2016); Bujack et al. (2017); Schloss et al. (2018).

C1  **Perceptual order**: colors in the color scale should follow a natural, intuitive progression

C2  **Perceptual uniformity**: the perception of difference between adjacent color levels should be uniform across each adjacent pair of colors

C3  **Discriminability**: colors in the color palette should be distinct and immediately discernible by most users, in both the map and legend

C4  **Appropriate associations**: color associations should be relevant to the data being mapped, based on culturally-specific associations reported in the literature

C5  **Accessibility**: colors should be readable and discriminable across the population, e.g., for users with CVD

### 2.2   Classification schemes: literature review and best practices

The seismic hazard values plotted on any hazard map take a continuous distribution. In order to map them using a discrete color palette, we need to select a classification scheme, or a procedure for classifying the continuous distribution into a set of distinct intervals that get assigned to a color. There are numerous approaches for this problem and we review the statistical and cartographic solutions in the literature.

    Earlier work in cartography has largely advocated for mapping continuous variables with a discrete set of colors, rather than a continuous color palette spanning the data range. Many authors have argued that it is more difficult for users to accurately



discriminate colors and read values off continuous maps (Dobson, 1973; Kennedy, 1994), though this has been debated in
the literature (Tobler, 1973; Muller, 1979). Still, discrete maps are ubiquitous for depicting natural hazards, especially for
public communication (Quinan and Meyer, 2015). The utility of discrete maps has been studied with perception tasks in user
experiments. In Padilla et al. (2016), participants saw elevation maps using either continuous or discrete classification schemes
and were more error-prone with the continuous map compared to the discrete maps, particularly for tasks requiring a steepness
judgment. Marti et al. (2019) also found that participants had complaints about the continuous color scale the authors used
in their seismic hazard maps and suggested that a discrete map would be more effective, as was echoed by Fyfe and Molnar
(2020).

    Classification schemes can thus lead to better reading and interpretation of maps. The first step for many classification
schemes is to consider the number of categories, and thus colors, that will be used in the map. Some authors have suggested
that between five and seven categories can be well-distinguished, with their meanings remembered (MacDonald, 1999; Miller,
1956), based on principles of level-of-detail management and cartographic generalization (Çöltekin et al., 2017). Still, the
effects of the number of categories in discrete maps on, e.g., hazard or risk perception has yet to be empirically studied. We
thus posit that the optimal number of categories is determined by the distribution and patterns of the data being mapped, as
well as other map-specific features. Because it is difficult for map users to discern too many categories, as described above, it
is critical that discrete maps use only so many categories as can completely convey the important patterns in the data.

    The map designer must also select between numerous available classification methods. The most basic approaches evenly
split the data distribution into either equal intervals or equal quantiles (that is, equal probability mass). While such approaches
may have the advantage of appearing conceptually straightforward, they ignore the characteristics of the data being plotted,
e.g., skewness, multimodality or spatial clustering. In particular, these schemes group together data into classes based on an
arguably arbitrary criterion (equal class width resp. equal number of data points for the equal intervals resp. quantile-based
schemes). This may distort the map, depicting patterns that do not exist in the data, rather than highlighting those that do
(Evans, 1977).

    Authors have proposed other solutions that split a dataset into classes based on more rigorous quantitative criteria, such that
values assigned to a class are more alike than those between separate classes. The most common classification approach used
today was proposed independently in the statistical (Fisher, 1958) and geographical literature (Jenks, 1967), and has become
widely adopted. Given a fixed number of classes, the Fisher scheme attempts to find the split of the data that minimizes the
variance of points within each class. The resulting breaks (also referred to as natural breaks and common across geographic
information systems) have optimally grouped data points together and split data points apart that are sufficiently dissimilar.
Other classification approaches attempt to, e.g., fit statistical parameters onto the dataset or prioritize spatial contiguity in the
resulting map; see typologies of classification methodology in Evans (1977) and Armstrong et al. (2003). Techniques also exist
for right-skewed data, such as the Head-Tails scheme (Jiang, 2013), where the dataset is first split at its mean into its "head"
and its "tail", which contains the data's long skew. The tail is again split at its mean and this procedure continues iteratively,
until some stopping point is reached (see examples in Jiang (2013) and Jiang et al. (2013)); the split points from each step of
the procedure (mean of that step's tail) comprise the class breaks.





The algorithmic classification schemes discussed above do not consider whether some data values may carry particular meaning, as they depend solely on the distribution rather than the context of the data. However, authors have discussed forming class breaks around meaningful values, e.g., using 50 percent as a class break when mapping voting results between two parties to symbolize the winning party, regardless of the distributions of voting results (Brewer and Pickle, 2002). Evans (1977) advocates that such "class limits will be used in the few instances where they are available." Though no literature exists

to compare such pre-selected meaningful class breaks against algorithmic alternatives, we posit that including breaks known to carry particular meaning can be beneficial to hazard communication.

### 2.2.1   Criteria for selecting classification schemes

Some previous studies in the natural hazards have compared different schemes based on different kinds of misclassification errors (e.g., Cantarino et al. (2019) compares Fisher, Head-Tails and several other schemes for landslide susceptibility maps).

To our knowledge, there is no previous literature on how classification schemes affect risk or hazard perception in users, neither for natural hazards nor other topics. This research gap means that, unlike colors, factors beyond experimental evidence must be used to select between classification schemes. Based on the literature surveyed, we propose the following three criteria as critical for selecting a classification scheme for hazard maps.

    L1  **Likeness**: the classes in the classification scheme should contain all data that are alike, based on a quantitative measure,
270        and should break apart data that are not alike.

    L2  **Sufficiency and completeness**: there should be enough (but not too many) scale breaks in order to be able to immediately
       see the primary patterns in the data.

    L3  **Signification**: scale breaks should ideally signify meaningful values or changes in the distribution, rather than arbitrarily
       selected values.

## 3   Application to German seismic hazard

In this section, we apply the criteria and concepts introduced in the previous three sections to redesign the German seismic hazard map presented in Fig. 1.

### 3.1   Hazard analysis and issues for Germany

Seismic hazard in Germany is spread across several regions, particularly the Hohenzollerngraben region (near Albstadt), the
Rhine valley (near Aachen), and in contours surrounding these regions across central-western and southwestern Germany. There is also an elevated hazard zone in eastern Germany, east of Chemnitz. Basics of the tectonic and structural geological rationale behind the development of the PSHA model are provided in Grünthal et al. (2018). Seismic building standards for Germany are required for ordinary structures (i.e., not those for essential infrastructure, e.g., dams or nuclear power plants, for which stricter standards apply) but only where seismic hazard is sufficiently high, i.e above a PGA threshold of 0.4 $\frac{m}{s^2}$. This



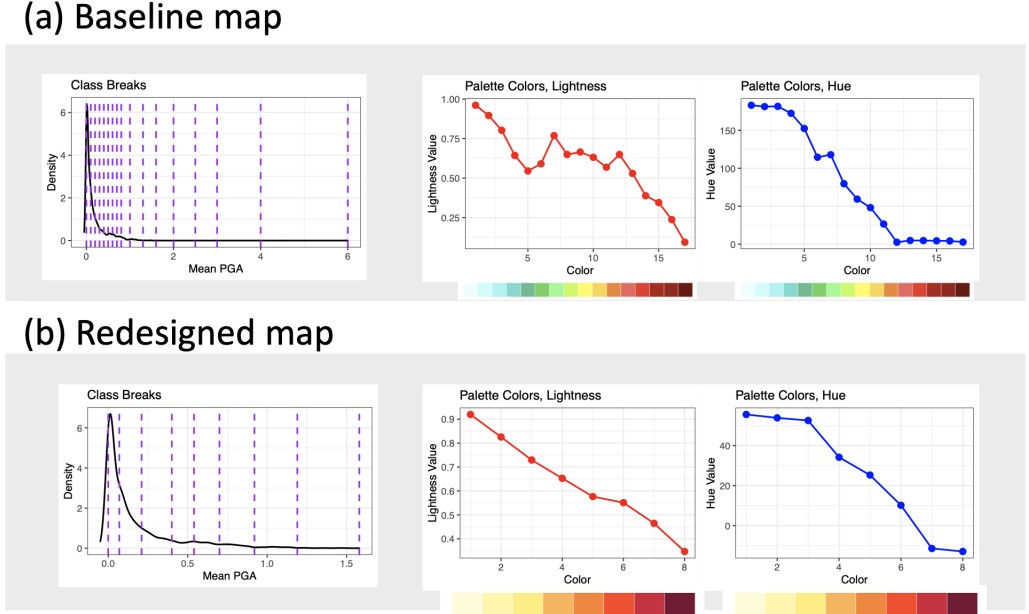

**Figure 2.** Classification schemes (left) and color palettes (right) for the baseline German seismic hazard map (a; see Fig. 1) and the redesigned version (b; see Fig. 4), following the criteria described in Sect. 2. Lightness and hue profiles indicate the extent to which the color palettes follow the criteria.

value thus became a critical threshold in our hazard map redesign (see section 3.2). Although it stems from a specialist user group, we posit it will be useful for all users to distinguish "non-significant" hazard (PGA < 0.4 $\frac{m}{s^2}$) from the rest of the hazard distribution.

### 3.2 Choosing the classification scheme

We first chose a classification scheme to represent the continuous distribution of mean PGA in a discrete map. The baseline
classification scheme (see Fig. 1) covers a larger range than the values plotted in the map, likely because the legend is intended to cover a wider span of the PSHA ensemble than just its mean (see Fig. 25 in Grünthal et al. (2018) that uses the same legend to represent a larger range of values). It is furthermore the legend used for an interactive web portal showing extreme percentiles from the German seismic hazard assessment (GFZ). As we are solely interested in a static single-map representation of seismic hazard (i.e., one that may be used in news media or reports for decision-makers), we limit our scale to only those values in the
mean PGA map.

The baseline map uses a modified equal-interval scheme with round decimals, where the interval increases with increasing PGA. While this may seem a straightforward approach, the breaks chosen do not meet the criteria listed in Sect. 2.2. In particular, they do not carve out similar hazard levels into separate classes using the underlying data (criterion L1); they are





moreover chosen based on other data not shown in this plot (lower and upper percentiles of the PGA distribution), rather
than the data being mapped (mean PGA, see Sect. 1.1). The baseline scheme also has 16 categories, which is far above the
literature-backed rule of thumb detailed in Sect. 2.2. Failing to consider sufficiency (criterion L2) in favor of including extra
detail comes with the additional need to assign the excess of classes to colors, many of which are difficult to distinguish from
one another (color criterion C3). Finally, these breaks fail to communicate meaningful hazard values (criterion L3) as they are
arbitrarily chosen round intervals, which are unrelated to the key thresholds of seismic hazard in Germany.

We developed a new classification scheme that meets our three pre-specified criteria for classification. We first split the
distribution of hazard values into "lower" and "higher" classes based on the critical value of 0.4 $\frac{m}{s^2}$, the hazard level at which
seismic building standards must be applied in Germany. The hazard values below 0.4 $\frac{m}{s^2}$ (the "lower" part) were split into three
classes and the values above 0.4 $\frac{m}{s^2}$ (the "higher" part) were split into five classes, for a total of eight classes. In each part,
we used the Fisher classification scheme (satisfying criterion L1), as implemented in the classIntervals function in the classint
package in R (Bivand et al., 2020). Our classification thus communicates a critical value that splits seismic hazard for Germany,
satisfying criterion L3. Unlike in the baseline map, the redesigned classification scheme was capped at the maximal value for
the mean PGA data being mapped (criterion L1).

    We also experimented with several other classification schemes. We first considered different numbers of classes, and created
maps with 5-9 total classes, split in several ways between the "lower" and "higher" parts of the distribution. We settled on the
split shown in Fig. 2, as this was the smallest number of classes in each part that sufficiently represented the spatial patterns of
hazard across Germany (criterion L2). This also aligns with property 4 of our communication goals regarding keeping spatial
patterns salient (see Sect. 1.1.1).

    Since seismic hazard follows a long-tailed distribution, we also considered the Head-Tails scheme on the "higher" part,
stopping the algorithm at the step when its tail did not contain at least 40% of the full set (see Fig. S2 in the Supplement). This
broke the high-hazard values into eight classes. Only southwestern Germany had a noticeable difference in the depiction of
hazard, as this is where the most extreme hazard is located. Fig. 3 shows that the Head-Tails scheme used many more classes
to depict this steep increase in hazard, relative to the Fisher scheme. As one of our communication goals is to avoid hazard
appearing concentrated in space (properties 1 and 2 of our communication goals), we opted against the Head-Tails scheme.
We also experimented with a quantile classification scheme (see Fig. S3 in the Supplement) as this was preferred by a single
empirical study we found [Brewer and Pickle, 2002] but opted against this as it flattened the distinction between high and
extreme hazard, violating property 3.

### 3.3   Choosing the color palette

We attempted to improve on the rainbow color palette, which is commonly used for seismic hazard (Fig. S4 in the Supplement
shows several other published seismic hazard maps, all employing a similar color palette). Despite the ubiquity of the rainbow
palette (Borland and Taylor, 2007; Westaway, 2022), scholars have long described why it is in fact "unscientific" (Crameri
et al., 2020), particularly for hazard maps. The hue and lightness profiles of Fig. 1's rainbow palette are given in Fig. 2 and this
sheds light on why it fails to meet the five criteria isolated in Sect. 2.1:





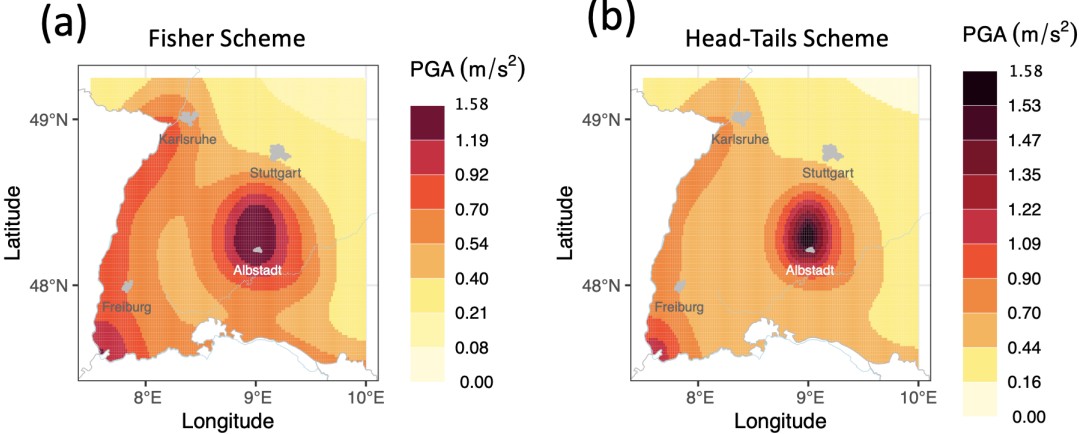

**Figure 3.** A closeup of southwestern Germany for the seismic hazard maps made using the Fisher class breaks (a; see Fig. 4 for the full map) and the Head-Tails class breaks (b; see Fig. S2 in the Supplement for the full map).

C1 Neither the lightness nor hue profiles are monotonic, meaning that it is not perceptually ordered

C2 The lightness profile is non-linear, meaning that it is not perceptually uniform

C3 Many of the colors, especially at the upper end of the scale, are non-discriminable, possibly because they have such a limited range of hues

C4 The palette uses blue and green shades that are commonly connoted with peaceful, tranquil and soothing associations in Western cultures, which are not appropriate for a hazard map

C5 The palette combines reds and greens, making it not CVD-accessible

The rainbow palette's lightness profile (see Fig. 2) leads to further problems, as pinpointed by other authors. When used in maps, this palette can cause the appearance of "false gradients introduced by a non-monotonic lightness profile, which accelerates at a different rate than the data it represents" (Thyng et al., 2016). The result is that map subregions may appear to be of nearly identical color and then sharply transition to another color. This can falsely be interpreted by viewers as a real gradient in the data, which "actively misleads the viewer by introducing artifacts to the visualization" (Borland and Taylor,
345    2007).

To address these issues, we selected a new color palette following the five criteria given in Sect. 2.1. We began with a Yellow-Orange-Red color scheme with eight colors (Yl-Or-Rd-8) from RColorBrewer, an evidence-based palette tool (Harrower and Brewer, 2003). This color palette is known to be accessible to people with the most common forms of CVD, as it does not combine red and green (criterion C5). Yl-Or-Rd-8 also has a monotonic progression of both lightness and hue, thus allowing
a single intuitive perceptual order (criterion C1); it can also be modified to optimize discriminability (criterion C3) and for





linear jumps in lightness and hue, creating perceptual uniformity (criterion C2). Furthermore, Yl-Or-Rd-8 has clear cultural connotations in Western cultures that begin with caution and preparation (yellows) and rise to danger and risk (reds) (criterion C4). This palette has furthermore been supported to depict different degrees of hazard by literature reviews (Bostrom et al., 2008), focus groups (Fyfe and Molnar, 2020), user experiments (Miran et al., 2017; Thompson et al., 2015) and guidelines for
science communication (Doore et al., 1993; Haynes et al., 2007).

We chose three shades of yellow for the first three colors (see Fig. 2). Lightness and hue progressed in even jumps from the lightest shade of yellow that was visible on a white background (as in the legend) to the darkest shade of yellow produced by the Yl-Or-Rd-8 color scheme. This was to associate the first three scale breaks with caution, as these correspond to the "lower" part of the distribution. For the next five colors (the "higher" part of the distribution), we made exactly even jumps in both lightness
and hue (criterion C2) between the final yellow color and a dark red color produced by the Yl-Or-Rd-8 scheme. We found that this led to several shades of red that were difficult to differentiate and we manually adjusted the color lightness and saturation until the colors were judged to be discriminable (criterion C3), while maintaining the same color order in both hue and lightness (criterion C1). We then modified the final color to diverge slightly from the Yl-Or-Rd-8 palette, in order to highlight how the extreme end of the hazard distribution differs from the high hazard preceding it (property 3 of our communication goals, see
Sect. 1.1.1). We chose a shade of dark brown that preserves perceptual order, uniformity and discriminability of the overall color scheme, while maintaining relevant color connotations and colorblind-friendliness.

### 3.4   Choosing the legend and annotations

Finally, we also experimented with different legend types and map annotations, which also have a (limited) base of empirical literature. The baseline map positioned its legend under the map, possibly because it was part of a set of three maps. In our
redesign, we follow traditional cartographic style and position the legend on the right side of the map, which Edler et al. (2020) has shown can lead to faster processing of the map. We follow Li and Qin (2014) in aligning the position of the color patches and labels and having a larger spacing between the label and its color patch than between color patches.

We further added several layers of annotation to the redesigned hazard map. We marked key cities in Germany, either with a land area above 65 km$^2$, or that were in important hazard zones. We marked each city with a polygon showing its boundaries
rather than a single point positioned at its geographical center. This followed internal evaluation (see Sect. 4 and Supplement S6) and spotlights how urban seismic hazard can span multiple hazard levels in many German cities. We added labels for each city, printed in black where the background color was sufficiently light, and white where the background color was too dark (Albstadt), following recommendations from Brychtova and Çöltekin (2016). We considered labeling contours corresponding to important hazard values but ultimately omitted these (see Supplement S6).

## 4   Evaluating redesigned seismic hazard maps

We tested the redesigned German seismic hazard map in a two-phase evaluation. We first created multiple prototypes that differed on certain features (e.g., the classification scheme or how cities were marked on the maps) and solicited structured





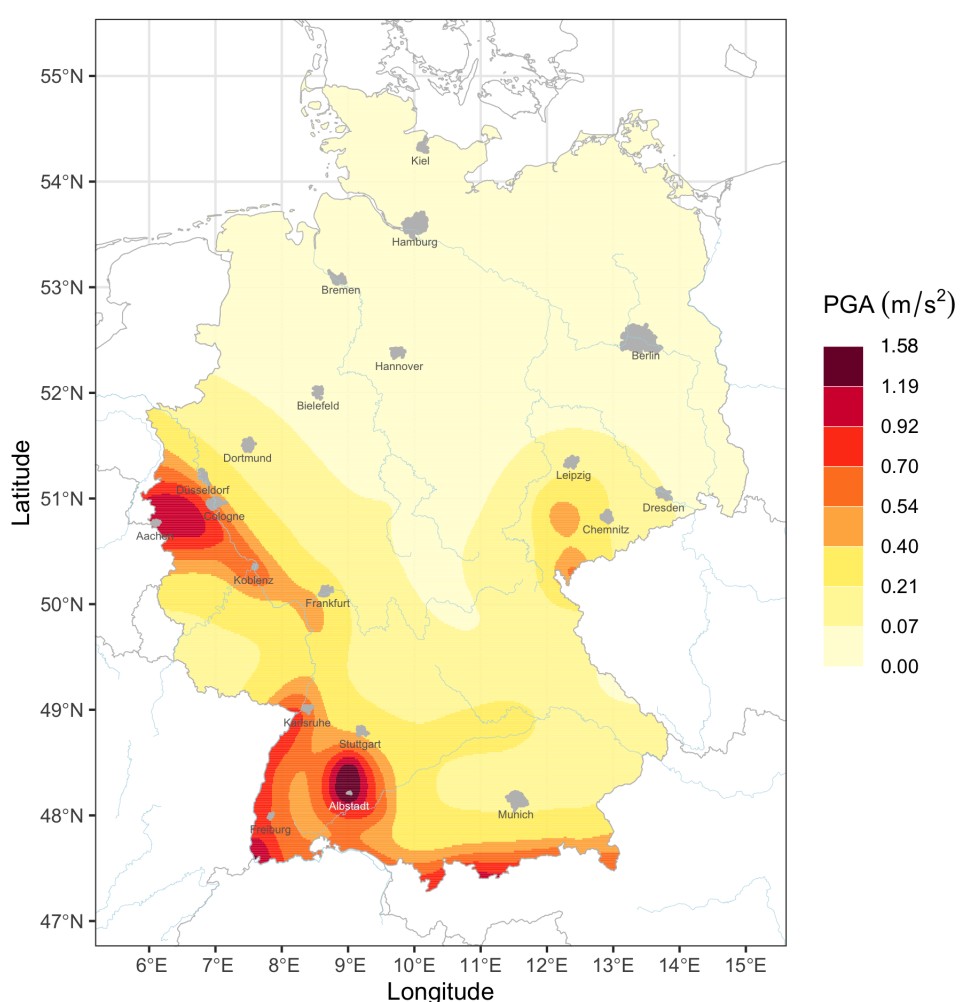

**Figure 4.** Redesigned German seismic hazard map fulfilling evidence-based criteria for color (see Sect. 2.1) and data classification (see Sect. 2.2).





feedback from seismic hazard experts. This is described in Supplement S6. The expert feedback helped us finalize a single redesigned map (see Fig. 4) which, in the second phase, we compared to the baseline map with a controlled user survey

focusing on the users' awareness of the four seismic hazard properties and one critical threshold.

## 4.1 Participants and design

For the user survey, we recruited early career researchers in the natural and engineering sciences from universities in Germany where the authors are affiliated. The participants had some expertise in geosciences or engineering but not necessarily in seismic hazard, which is similar to many of the technical users of hazard maps. We developed the survey in the survey platform

Qualtrics and then invited these researchers to evaluation workshops, in which they completed the survey and we then presented the research behind the redesign. Invited researchers were also able to take the online survey at a later time during a fixed period. 43 individuals began the survey and five did not complete questions for both maps, meaning they were excluded from analysis. Another participant did not provide responses to the open-ended and demographics questions, but was still included in the other analyses. The final sample included 38 participants (42.1% female, median age of 30).

We pre-registered the survey and analysis plan in the Open Science Framework (https://osf.io/puerc/?view_only=2b747decfbfb4093a9e925e5fe09cd48, last access: 10 October 2022). The study was approved by the Institute for Advanced Sustainability Studies (IASS) ethics ombudsperson following the IASS procedure for ethics approval.

Participants began by reading a basic explanation of PSHA and what seismic hazard maps show. They were asked to use

only the given map to answer the question, rather than any other maps or background knowledge. After providing informed consent, they answered map-reading and hazard perception questions first with one map and then with the other (within-participants design), with the map order randomly assigned. We developed two equivalent versions of each map-reading and hazard perception question, and the question version was also assigned randomly so that no question would be repeated on both maps.

Map-reading questions asked participants to read off the maximum hazard level for a given city. Participants answered three of these questions for each map, and their responses were scored for accuracy. Hazard perception questions were designed around the four hazard properties (see Sect. 1.1.1). For each property, we posed a statement about a city or pair of cities on the map and asked for the participant's level of agreement, on a seven-point scale with the following labels: "Strongly disagree", "Disagree", "Slightly disagree", "Neither agree nor disagree", "Slightly agree", "Agree", "Strongly agree". Cities were chosen

such that agreement with the statement indicated an awareness of that hazard property. For example, for property 1 (seismic hazard is not spotty, it changes continuously over space), participants marked their agreement with the statement: "Damaging shaking may be expected in Düsseldorf, according to the map" (Düsseldorf is in a zone of moderate hazard, contouring off a high-hazard zone near Aachen and thus, could expect damaging shaking). The full set of evaluation questions is provided in Supplement S7.

We also asked for participants' perception on whether particular cities needed seismic-resistant design for ordinary buildings. The aim was to see whether our redesign could sufficiently communicate the critical threshold of 0.4 $\frac{m}{s^2}$ without needing to





provide this explicitly. We asked about two sets of cities: those we expected to be simple (at extreme ends of the scale) or difficult (in the middle of the scale).

Since there are no "correct answers" to these perception questions, we simply analyzed the within-participant differences in
response between the two maps to see if either map improved hazard awareness. Our inferential analysis for both map-reading and hazard perception compared these map differences (within-participant) to zero, using the Wilcoxon signed rank test, a non-parametric extension of the paired-samples $t$-test. Given the discrete nature of this data, we used the Pratt correction for ties [Pratt, 1959]. We set the significance threshold to 5% and accounted for multiple testing with the Bonferroni correction.

After completing the map-reading and hazard perception questions for both maps, the participants answered three open-
response questions. We asked for feedback on their preference between the two maps and reasoning behind this, what they would change about either map to make it more useful and which evaluation questions they found difficult. We used a systematic and intersubjectively verifiable approach to qualitatively analyze participant responses. After individually reading through all responses, we identified a set of distinct themes that could characterize the content in each response. We allowed each response to be categorized by one or more themes and studied the frequency and meaning of the theme categories. Finally, we asked
for participants' demographics (age, gender, nationality) and academic background (university, discipline of study) to explore potential relationships between these and map effects.

## 4.2 Evaluation results

The first set of questions targeted how accurately participants could read values off the two maps. The majority of participants could accurately read most cities off both maps (see Fig. S5 in the Supplement), with participants averaging 71.9% correct with
the baseline and 77.2% with the redesigned maps. The maps were statistically indistinguishable in how accurately participants read them ($p$=0.4835).

We then estimated how each map affected perception of the hazard properties, see Fig. 5. The first property was that seismic hazard is not spotty but rather changes continuously in space. There was greater agreement with the statements asked for this property for the redesigned map (81.6% "in agreement", or selecting one of the three "agree" options), than for the baseline
map (36.8% in agreement). The 95% confidence interval (CI) for the difference in response between the redesigned and baseline map was entirely positive, meaning that the redesigned map significantly improved awareness of this hazard property ($p$=0.000019). The second property was about seismic hazard being possible even in areas where hazard is not mapped as high. We again found that participants agreed with the corresponding statements more for the redesigned map (89.5% in agreement), than for the baseline map (60.5% in agreement). The 95% CI for the participant-specific difference between maps was always
positive, so participants showed significantly ($p$=0.0013) more awareness of this property using the redesigned map.

The differences between the maps were smaller regarding perception of the other hazard properties. Both maps facilitated understanding of the third property (extreme hazard differs from high hazard), with the vast majority responding "Agree" or "Strongly agree" across both maps. The 95% CI for this difference contained zero, meaning the difference was not statistically significant ($p$=0.014), when accounting for multiple testing. Both maps also equivalently conveyed the fourth property (spatial





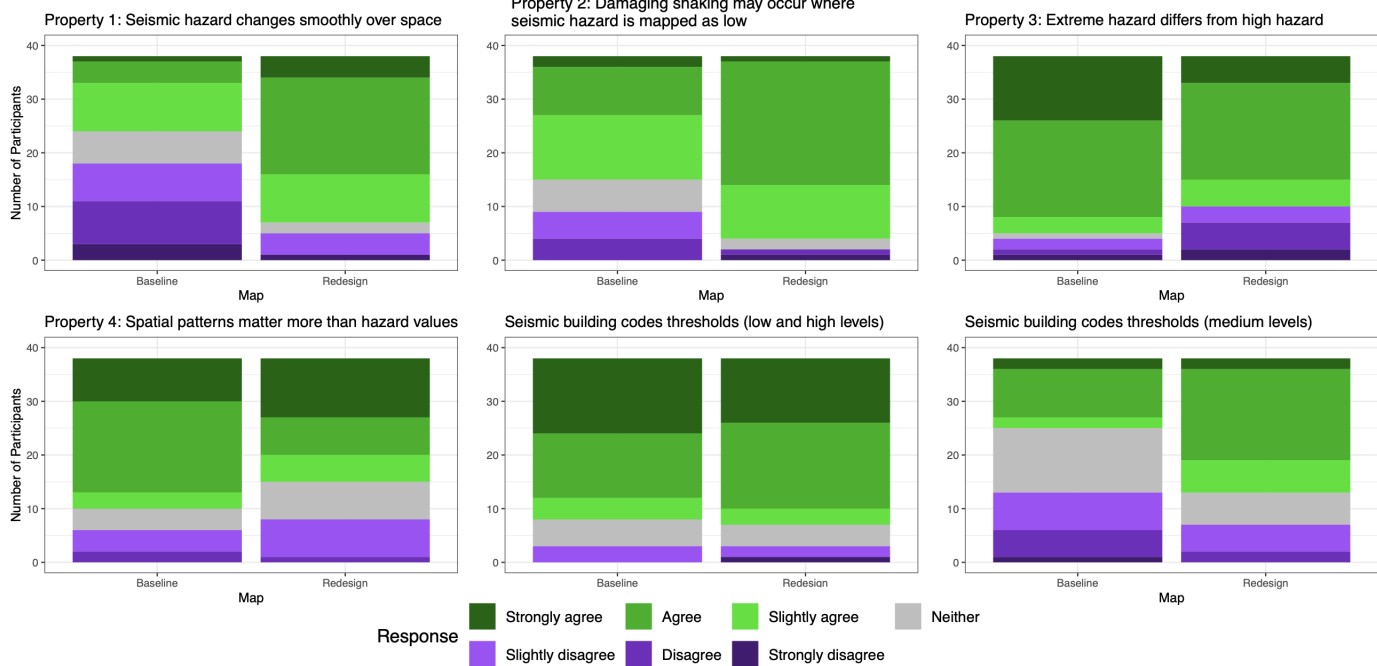

**Figure 5.** Participant responses for hazard perception questions, under each map. Questions related to each hazard property (see Sect. 1.1.1) and the seismic building code threshold. Agreement responses (colored with greens) indicate awareness of the hazard principle/threshold. See Supplement S7 for the set of hazard perception questions.

patterns of hazard are more meaningful than the hazard values), meaning that the difference in response for these questions was not statistically significant ($p$=0.53).

We also asked several questions about whether certain cities were perceived to need seismic building codes due to their hazard level. The difference between maps was negligible for cities where this question was more straightforward (Albstadt and Leizpig, both at extreme ends of the scale). Both maps led participants to correctly perceive that Albstadt requires building
codes and Leipzig does not (78.9% of participants in agreement for the baseline map and 81.6% for the redesigned map), so the 95% CI for this map difference crossed zero ($p$=0.76). For cities where this distinction was less clear (Freiburg, which requires building codes; Frankfurt, which does not), there was a slight difference between maps. The redesigned map led participants to more often make interpretations consistent with the actual building code (65.8% consistent with the building code), compared to the baseline map (34.2% consistent), though the within-participant difference was marginally insignificant ($p$=0.046) when
accounting for multiple testing, with its 95% CI containing zero.

We then qualitatively analyzed feedback questions, seeking to identify the recurring themes across responses. The first feedback question asked participants for their preferred map and the reasons why. While 52.6% of the participants preferred the redesign map, 31.6% preferred the baseline and 15.8% did not indicate a single preference. Despite different preferences, color features (discriminability, perceptual uniformity, etc.) were most commonly mentioned by both groups (see Table 1). For





| Theme | Baseline | Redesign |
|---|---|---|
| Color features (discriminability, perceptual uniformity, etc.) | 50.00 | 37.04 |
| Communication of hazard and risk (e.g., critical hazard values) | 3.85 | 18.52 |
| Communication of high/extreme hazard levels | 0.00 | 11.11 |
| Color associations/conventions | 23.08 | 7.41 |
| Issues with data classification | 7.69 | 11.11 |
| Issues with upper end of data scale | 11.54 | 7.41 |
| Other | 3.85 | 7.41 |
| Total | 100 | 100 |

**Table 1.** Themes referred to by participants when describing their preference between the baseline and redesigned maps and the proportion of responses that mentioned each theme.

example, a participant who preferred the baseline map found it "easier to discriminate between different colors than different tones of red" and a different participant who preferred the redesigned map said "it more clearly implies a continuous increase in danger". Participants who preferred the redesign also reported a greater range of reasons, e.g., commonly addressing either critical or high/extreme hazard levels. For example, one participant found that with the redesign, "it is easier to identify where the most exposed areas are located."

Responses to the other feedback questions were more similar between participants that preferred one map over the other. Regarding which aspects of the map could be improved, participants commonly brought up classification breaks needing to be meaningful or interpretable; for example, a participant asked "which PGA could really affects constructions [sic] or be of potential threat to anyone/anything?". Participants also mentioned that the double legend labels were difficult to interpret. There was an even spread in issues participants reported having with the survey, though 47.4% of participants reported no issue

with the survey. Problems that were mentioned by multiple participants included not understanding thresholds corresponding to the building code questions and or wording in some questions, such as what "damaging shaking" means.

We found that demographic and academic characteristics largely did not influence any of the reported map effects. Due to the small sample size, we examined these characteristics as binary variables, splitting participants by gender (male or female), age (younger or older than the median age of 30 years), modal nationality (German or non-German), modal university (attending

University of Potsdam or not) and the two modal academic disciplines (geosciences or engineering). Map-reading accuracy and tendency to agree with the hazard perception questions were not qualitatively different in any of these groups, though sample sizes were too small to detect small differences. There was a slight difference in map preference between some subgroups (see Table 2) in particular for academic discipline, where geoscientists tended to prefer the baseline map and engineers the redesigned map.





|  | Baseline | Both | Redesign |
|---|---|---|---|
| Female ($n = 16$) | 37.50 | 6.25 | 56.25 |
| Male ($n = 20$) | 25.00 | 25.00 | 50.00 |
| Younger (Age $< 30$, $n = 17$) | 35.00 | 25.00 | 40.00 |
| Older (Age $\geq 30$, $n = 20$) | 29.41 | 5.88 | 64.71 |
| German ($n = 9$) | 22.22 | 22.22 | 55.56 |
| Non-German ($n = 27$) | 37.04 | 14.81 | 48.15 |
| University of Potsdam ($n = 19$) | 26.32 | 26.32 | 47.37 |
| Not at University of Potsdam ($n = 19$) | 36.84 | 5.26 | 57.89 |
| Geosciences students ($n = 12$) | 50.00 | 16.67 | 33.33 |
| Engineering students ($n = 21$) | 23.81 | 14.29 | 61.90 |
| All ($n = 38$) | 31.58 | 15.79 | 52.63 |

**Table 2.** Preferences between the baseline and redesigned maps, split by subgroups on the person-specific variables. Some participants responded that they liked both maps, without a clear preference between them. Note: two participants did not specify their gender and five participants did not identify as a geoscience or engineering student.

## 5 Discussions and conclusions

### 5.1 Systematizing the design of hazard maps

Previous work on risk communication for seismic hazard maps has largely focused on collecting user feedback on different map designs, while citing non-specific "best practices" in describing how design choices were made. We sought to disentangle what graphical criteria actually constitute these best practices in map design, particularly focusing on color palettes and data classification, which have been prioritized by previous authors (Marti et al., 2019; Fyfe and Molnar, 2020; Thompson et al., 2017). The aim was not to provide a recipe for how to design hazard maps, but rather a criteria-based framework around which to make design decisions. We surveyed the color and visualization literature, using the Hue-Saturation-Lightness color model to elucidate three central criteria (perceptual order, perceptual uniformity, and discriminative power) found to be key to correct map interpretation, based on decades of empirical experiments. We also found a body of empirical literature supporting choosing colors with psychological associations relevant to the map topic as well as those that are colorblind-accessible.

We then surveyed the cartographical literature to identify different data classification methods that could be relevant for hazard maps. Traditional approaches such as equal-interval or quantile classification can obscure or even distort the true features of the data being mapped. We reviewed classic and modern approaches designed to find class breaks that fit the data. We also discussed the research behind using a sufficient but not excessive number of class breaks and making them meaningful when possible. As empirical evaluations to compare classification schemes are scant, we propose three general criteria that have support from the literature.





We implemented these criteria to select a new color palette and classification scheme for the German seismic hazard map. The classification scheme was split at the PGA of 0.4 $\frac{m}{s^2}$, the critical threshold at which Germany requires seismic building codes for ordinary buildings. The lower part of the distribution was split into three classes and the higher part was split into

five classes, using the Fisher classification scheme, which well-represented the key features of the data, grouping data into classes that were quantitatively similar. The map used a carefully designed and colorblind-friendly yellow-orange-red-brown color palette. The lower part of the scale used shades of yellow associated with caution, whereas the higher part used red colors associated with risk and danger, to appropriately convey the meaning of each part of the scale. Colors were optimized to be perceptually uniform and discriminable, with a single intuitive and identical ordering by both lightness and hue.

## 510    5.2    Assessing the redesigned German seismic hazard map

We tested the two German seismic hazard maps on a sample of early career researchers using a survey with map-reading and hazard perception questions. Participants could read both maps equally well, contrary to the suggestions of Gaspar-Escribano and Iturrioz (2011) that classification schemes with unequal intervals come at a cost to map-reading. While participants were only around 74% correct across the two maps, open-ended responses indicated that multiple participants struggled to under-

stand the double legend labels. Indeed, most of the map-reading mistakes (85.3% for the baseline map and 84.6% for the redesigned map) came from selecting the adjacent value above or below the correct one. Mistakes were also more common when cities spanned multiple colors, with participants often incorrectly choosing the lower hazard level, potentially indicating that they failed to notice that part of the city was in a higher hazard level; this appeared to affect both maps equally. Contrary to the suggestions of the expert evaluation (see Supplement S6), double legend labels and cities marked with polygons may be

difficult to read precisely, even for a technical audience.

The redesigned map was successful in generating awareness of each key property of seismic hazard, as well as the critical threshold. Across the six question types, participants were 60.5-89.5% in agreement with the statement (selected either Slightly agree, Agree, or Strongly agree, and 47.3-73.7% of participants selected Agree or Strongly agree), indicating their abundant awareness of the hazard property. Relative to the baseline map, the redesign improved perception of two closely-related prop-

erties of seismic hazard: (1) that it spreads continuously over space and (2) that hazard can occur in areas even if they are not marked high on the map. On the other hand, both maps equally communicated the two other framing properties, that extreme hazard differs from high hazard and that spatial patterns are of primary importance. Both maps communicated which cities required seismic building codes roughly equally, both for cities at the far end of the scale (that would arguably be easier to answer for), and in the middle of the scale. These results held across the sample, regardless of demographics or academic

background, indicating they may be robust to these user characteristics for similar technical audiences.

There are several aspects of the redesigned colors and classification schemes that may explain these evaluation results. The two properties where the redesign outperformed the baseline both have to do with the spatial diffusion and continuity in the regional patterns of seismic hazard. Users had awareness with the redesigned map, but not the baseline, that damaging shaking could be expected in cities that were close to, but not directly within zones of higher hazard (e.g., Karlsruhe, Koblenz,

and Düsseldorf). The redesign's classification scheme groups hazard values more objectively and into fewer classes than the





baseline, thus avoiding transmitting too much information (Marti et al., 2019). This may have made it easier to identify these cities as having a medium (and sufficiently important) hazard level. Furthermore, these classes were orange in the redesign and blue or green in the baseline; as found in multiple studies (Fyfe and Molnar, 2020; Miran et al., 2017; Thompson et al., 2015), orange may be better associated in Western cultures with damaging hazard than blue/green.

Responses to the open-ended questions indicated that there was a slight preference towards the redesigned map but that participants used many of the same explanations to describe preferences for either map. In particular, participants referred to the ability to discriminate colors and order them uniformly for both maps. Participants mentioned psychological aspects of the color to describe their preferences both for the baseline and redesigned map, in contrast to Fyfe and Molnar (2020)'s suggestion that evocative colors may not be preferred by map users. Participants also expressed the importance of classification breaks that

signify a particular meaning. It is clear that the visualization elements (color, classification breaks) around which our redesign was structured were pertinent to these map users, both for their preferences and hazard perception between the maps. While these results were consistent across demographic subgroups, there was a preference for the baseline map only for geoscience researchers (see Table 2), potentially due to the greater ubiquity of the rainbow color scale in this field (Westaway, 2022).

### 5.3 Practical implications and limitations

Our criteria-based design method and application to German seismic hazard maps suggest several practical implications for designers of future hazard maps. First, it is possible to isolate empirically-evidenced criteria for color and classification in order to systematize best practices for hazard mapping. We recommend future hazard map designers to follow these criteria to generate candidate maps that are supported by the visualization and cartography literature. To decide between candidate maps, it is helpful to pre-specify a list of communication goals, or properties of the hazard process that are essential to communicate.

Finally, it is critical to evaluate the redesigned map with formal user testing, in order to measure whether the map's goals were effectively communicated.

Our evaluation suggested that the specific design choices we made (the Fisher classification scheme together with our customized yellow-orange-red-brown color palette) produced a map that was read sufficiently accurately and succeeded in users perceiving all communication goals. In two of these goals regarding regional spatial spread of seismic hazard, the redesigned

map outperformed the initial map, which used a suboptimal color palette and classification scheme; the redesigned map was also slightly more preferred over the initial map. Hazard maps adhering to evidence-based criteria for colors and classification schemes, and designed with key communication targets in mind, can reach these targets better than maps designed without such a framework. While our design choices may not necessarily be appropriate for other hazard types (i.e., where the hazard values' skewness is a priority to communicate), the criteria-based procedure we outline for designing a hazard map is.

Our study presented some limitations that can be addressed in future work. Our isolated criteria are not comprehensive and other considerations may be important in effective hazard map design. Visualization research suggests that the spatial frequency of patterns in a map may guide which color palette is optimal for it (Reda et al., 2018); that is, high-frequency hazards (e.g., landslides, which are concentrated in space along sharp slopes) may require different color palettes than lower-frequency hazards (e.g., seismic hazard, which spread over larger areas). Future redesigns can address these findings and build further



bridges between the natural hazards and visualization research communities. Furthermore, our evaluation study was limited in both its sampled population (early career researchers in Germany), number of questions (just one per communication goal, per map) and number of maps tested (a single redesign combining both color and classification improvements). Future work should disentangle the effects of color and classification changes through multiple maps, using a sample from a more general population and with an evaluation survey with more questions per communication goal.

*Code and data availability.* All code and seismic hazard data needed to produce the different hazard maps are available at https://osf.io/puerc/?view_only=2b747decfbfb4093a9e925e5fe09cd48.

*Author contributions.* We use the CASRAI Contributor Roles Taxonomy to categorize author contributions.

**Conceptualization**: MS, FC. **Resources** (developing hazard maps): MS, FC **Methodology** (method development and evaluation): MS, FC, PJS. **Investigation** (evaluation data collection): MS, PJS. **Formal Analysis** (evaluation data analysis): MS, PJS. **Writing - original**
**draft**: MS. **Writing – review & editing**: FC, PJS. **Supervision**: FC, PJS.

*Competing interests.* The authors declare that they have no conflict of interest.

*Acknowledgements.* We gratefully acknowledge Christian Bosse, Gottfried Grünthal, and the other colleagues and staff of the German Research Center for Geosciences that supported this work.



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
