# Peer review of "Criteria-based visualization design for hazard maps"

_EGUsphere, 2022_

## Author Comment (AC1)

Author Response to Reviewers for egusphere-2022-1064
Max Schneider, Fabrice Cotton, Pia-Johanna Schweizer

*Reviewer 1:*

*The manuscript presents design criteria for improving the communication of earthquake hazard maps by carefully considering colour and classification schemes. They apply these considerations to redesign a map of earthquake hazard in Germany. They evaluate the effectiveness of the redesigned map with a small survey, finding the new map improves users understanding of several important concepts related to earthquake hazards. They conclude with a discussion about applying these design criteria to other hazard maps.*

*This topic is relevant to NHESS, the manuscript is very well written with clearly defined objectives and conclusions that are supported by the results. One general comment is the scope is relatively narrow to a single map, and it would be interesting to see broader implications of how these design criteria could improve other hazard maps (e.g., low-probability high-consequence scenarios, regions with different hazard patterns). However, the authors clearly identify their scope, and their discussion section offers some insights into how these criteria could apply to other scenarios. Overall, this is a high-quality manuscript and I recommend it for publication with some minor revisions.*

We thank the reviewer.

**Specific comments**

- **Scope and other applications.** *One topic that would be interesting to further explore is the mapping of low-probability high-consequence scenarios, which are a common challenge across all natural hazards. The introduction begins by discussing this scenario and the typical right skewness of earthquake hazard, but the remainder of the paper narrows the focus to a single hazard variable (i.e., 475-year return period). It would be interesting to return to this topic in the discussion with some insights about what types of communication goals or colour and classification criteria could be helpful to communicate low-probability high-consequence events. Line 564 briefly mentions communicating skewness and I wonder if this could be expanded.*

We agree with this comment and will expand the sentence given in line 564. We can discuss how in situations where the skewness of seismic hazard is a priority to communicate (i.e., a communication goal), the Head-Tails classification scheme could be chosen over the Fisher scheme. Under this communication goal, the Head-Tails scheme can be argued to better classify the hazards values.

- **Capping the classification at the maximum value.** *While I understand how capping the classification with the maximum value shown on a map will optimize the communication goals on a single map, but I think there are limitations to this approach that are worth acknowledging. First, this method could distort the understanding of*

*extreme hazard if the highest values always reach the highest end of the scale, because in some datasets the maximum hazard may only result in light or moderate ground shaking. If a communication goal is understanding the severity of extreme hazard, then I think there is value in having some external constraint to avoid every map from appearing to have extreme hazard. Another concern is that fitting classification schemes to individual datasets could diminish communication for users who need to compare hazard in different scenarios, such as different regions or different return periods. In this case, it would perhaps be more meaningful to combine the data from all the maps in the analysis and create a standard classification to this data using similar methods done for the single map.*

The reviewer argues that capping each map with its highest value may lead to an interpretation that any region being mapped has extreme hazard, even for regions where "the maximum hazard may only result in light or moderate ground shaking". We do not disagree with the reviewer; indeed, the highest end of the scale is not the same between a 475 years vs. 2500 years return period or a PSHA map in Turkey vs. Germany. Developing a standard classification harmonizing the colors among maps developed for different return periods/regions is an interesting idea but such a goal is challenging and not one of our paper's objectives. Indeed, our key working hypothesis is that hazard maps should be developed according to clear communication goals. These communications goals and safety goals may also vary from one country to another and it is therefore difficult to impose a standard classification (it would mean standard and agreed-on communication goals and safety levels).

We believe that specifying "some external constraint to avoid every map from appearing to have extreme hazard" is problematic, because it is non-trivial to find such an external constraint, and due to the non-unique definitions of "high" or "extreme" hazard, this would be designer-dependent and thus not part of a systematic approach to hazard map design. We rather suggest to develop harmonized hazard maps at the global scale or at least continental scales (in parallel to maps developed at the national scales). Such maps may not have the resolution achieved at the national/regional level but can better evaluate the relative hazard levels from one country to another. For such maps, the communications goals are clearly defined and not imposed. Such efforts are developed at the EU level by the EFEHR Consortium (e.g. Danciu et al., 2022).

We appreciate the reviewer's point that there is a difference between "extreme ground shaking" and the level of shaking that can be found in moderate-hazard seismic regions like Germany. However, due to the right-skewed nature of seismic hazard in any region, there will always be a (small) zone of highest hazard. Since one of our communication goals is that this area (even in moderate-hazard seismic regions) will have larger impacts on expected losses than the area of next-highest hazard (lines 66-67), it must be well-distinguished from the preceding hazard class. We believe that the use of the phrase "extreme hazard" may have been understood to physically mean "extreme ground shaking", which is not what was meant (nor what is needed for the argument above), and we propose to change this phrasing to "highest hazard", both in line 66, and throughout the manuscript.

The reviewer additionally argues that classification schemes fit to the map's data can diminish communication when users need to compare hazard across maps. We acknowledge the challenge of designing hazard map classification schemes that can support comparisons across different regions or different return periods. The scope of our manuscript is restricted to single-map representations of hazard, as specified on line 52. However, the reviewer raises important points about practical uses of hazard maps (needing to compare across different regions or different return periods) that are indeed limitations of the method we propose. We will outline this in Section 5.3 on "Practical implications and limitations."

Danciu L., Nandan S., Reyes C., Basili R., Weatherill G., Beauval C., Rovida A., Vilanova S., Sesetyan K., Bard P-Y., Cotton F., Wiemer S., Giardini D. (2021) - The 2020 update of the European Seismic Hazard Model: Model Overview. EFEHR Technical Report 001, v1.0.0, https://doi.org/10.12686/a15

- ***Hazard level terminology.*** *Are there standard definitions for extreme, high, moderate, and low earthquake hazard? Are these relative terms, or does it directly relate to levels of peak ground acceleration for a set return period? It seems these terms are loosely defined in the manuscript and could be clarified. If one of the communication goals is to understand the difference between extreme and high hazard, it would help to know what this precisely means. Clarifying these terms would also help readers who are less familiar with earthquake hazards (including myself), since the only value provided for reference is the building code threshold of 0.4 m/s2. If there are objective definitions of extreme hazard, then these may also be meaningful thresholds to include in the classification. One specific place this impacts understanding is in lines 318-326 where the distinction between high and extreme hazard is used to compare the Fisher, Head-Tails, and quantile classification methods.*

No standard numerical definitions exist to differentiate between extreme, high, moderate, and low earthquake hazard. These are relative terms that may mean different things in different seismic regions. However, one way of avoiding the disconnect between our phrasing of "extreme hazard" and the idea of "extreme levels of ground shaking" is to simply replace "extreme hazard" with "highest hazard" throughout the manuscript, as proposed above. We will do this throughout lines 318-326 and elsewhere in the manuscript.

- ***Keeping track of goals and criteria.*** *When first reading the manuscript, I was slightly confused by the various lists of goals and criteria, since there are research questions, communication goals, and two sets of criteria. The criteria were easy to track with the C1-C5 and L1-L3 labels, but it was a little more difficult to reference back to the communication goals. Also, I found the current structure slightly confusing because the communication goals were presented before the research goals, which made the objective of the study unclear when reading. I wonder if some minor restructuring, short explanations, or labelling could make this easier to follow.*

We thank the reviewer for these helpful comments. We will change the numbering of the communication goals (lines 60-69) to start with a G (i.e., G1, …, G4). This will be changed

throughout the manuscript. The communication goals are related directly to the German seismic hazard map, which is a case study that illustrates the more general problems that the rest of the section introduces. As such, they are best positioned at the end of that section (Section 1.1.1), rather than after the research goals, which are given in Section 1.4.

- *I appreciate the high quality of the figures and tables. They are clear and simple to understand and support the main arguments of the manuscript.*

We thank the reviewer.

***Technical comments***

- *Line 276: Perhaps be more specific than "previous three sections" since there are only two previous sections that contain several levels of subsections.*

We will change the words "previous three sections" to "Sections 2.1 and 2.2" in this line.

- *Sect 3.3: The justification of the Yl-Or-Rd-8 colour palette is very strong in this section! It would be nice to see this colour scheme applied more broadly for all natural hazard communications.*

We thank the reviewer.

- *Line 363-365: What colour property was modified to change the dark red to brown? The argument about preserving perceptual order, uniformity, and discriminability is clear, but I am curious whether it was the hue, saturation, or lightness that was modified to make the extreme end of the scale distinct from the dark reds.*

We lowered the hue variable of the final color to change it from dark red to brown.

---

## Author Comment (AC2)

Author Response to Reviewers for egusphere-2022-1064
Max Schneider, Fabrice Cotton, Pia-Johanna Schweizer

*Reviewer 2:*

*The manuscript of Schneider et al. addresses the problem of the correct representation of seismic hazard models, to give the public the right information on the hazard levels at the various sites. The problem is deeply felt by those who produce seismic hazard models and I find it is important that it is tackled in rigorous scientific terms. I'm talking as a researcher involved in seismic hazard model elaborations and in their dissemination and the reading of the manuscript was very pleasant and flowing.*

*The work is very well organized and reaches the result after very serious analyses, also including tests with possible users.*

*I believe the manuscript can be published with sligthly revisions.*

*Regarding the detail of the structure of the article, I find the bibliographic research on the choice of the right colors very interesting, based on studies also carried out in different disciplinary fields.*

*I have no critical observations to make about the manuscript, except the constant reference to the criteria defined in sections 1 and 2, which require you to scroll back and forth through the pages.*

We thank the reviewer.

*What in my opinion needs an explanation to the reader, for the work to be usable in other countries as well, is the problem of the limited number of classes and the definition of class intervals. The German model proposes very low values, if compared with those of other European nations. In Italy the map with 10% of probability of excess reaches 3 m/s2, double the maximum value in Germany. In Greece and Turkey there are higher values. If we then take into consideration estimates of 2% in 50 years or estimates of spectral accelerations, the values are much higher, even up to 20 m/s2, an order of magnitude higher. Since the authors never mention this issue, I would like to know if the authors plan to use for any output the same color palette and modify the classes, or to add more classes; in the first case it would be impossible to compare maps for different return periods or for different spectral periods and the information that there are higher values in one map than in another would be lost. I think this information will be useful for many future users of the method, myself included.*

We agree with the reviewer's argument that different regions would require different classifications. To answer the reviewer's question (do we "plan to use for any output the same color palette and modify the classes, or to add more classes"?): we do not propose to use the same color palette for a different seismic region or map. Rather, we propose to select a classification scheme using the three criteria given in lines 269-274 and a corresponding color palette using the five criteria given in lines 205-212. The point of our method is that, following

these criteria for selecting classifications and colors, map designers can create maps to fit the patterns of hazard in their map.

*Another question is whether the 3 classes used for values lower than 0.4 m/s2 are not too many, to the detriment of a higher resolution of the higher values, for which I personally would prefer a greater number of classes. Personally, I am convinced that up to 10-12 classes are better for covering wider range of values.*

We understand the reviewer's argument that "three classes used for values lower than 0.4 m/s2 [may be] too many." The motivation behind this choice, as described in Section 3.2, was to appropriately communicate the dynamics in the "lower" region of seismic hazard (or areas with hazard < 0.4 m/s2). We discuss in the manuscript (lines 313-318) our experimentation with different classification schemes, which included 5-9 total classes, split in several ways between the "lower" and "higher" parts of the distribution. We explain that our classification scheme was chosen because it was the smallest number of classes in each part that sufficiently communicated the spatial patterns of hazard across Germany, which corresponds to criteria L2. That is, three classes was judged to be the smallest number needed to show the patterns within the "lower" zones of seismic hazard in Germany. Having more classes (i.e., in the "higher" part of the distribution) would be possible, but violates the numerous research articles that conclude that humans have difficulty discriminating many colors off maps (e.g., Padilla et al (2016); Cöltekin et al. (2017); Miller 1956; MacDonald 1999). There is a lack of research on what the ideal number should be and so we have posited that it should be based on using only so many classes as can completely convey the important patterns in the data (leading to criterion L2). This is described in detail in lines 228-235.

*Figure 3b has a different legend than those reported in the supplement as Figure S2. Maybe I misunderstood the caption of figure 3?*

We thank the reviewer for noticing this discrepancy and will update the legend in Figure S2. Figure 3b has the correct legend.